# Can You Trust Your Model's Uncertainty? Evaluating Predictive Uncertainty Under Dataset Shift

**Yaniv Ovadia**[*]
Google Research
yovadia@google.com

**Emily Fertig**[*][†]
Google Research
emilyaf@google.com

**Jie Ren**[†]
Google Research
jjren@google.com

**Zachary Nado**
Google Research
znado@google.com

**D Sculley**
Google Research
dsculley@google.com

**Sebastian Nowozin**
Google Research
nowozin@google.com

**Joshua V. Dillon**
Google Research
jvdillon@google.com

**Balaji Lakshminarayanan**[‡]
DeepMind
balajiln@google.com

**Jasper Snoek**[‡]
Google Research
jsnoek@google.com

## Abstract

Modern machine learning methods including deep learning have achieved great success in predictive accuracy for supervised learning tasks, but may still fall short in giving useful estimates of their predictive *uncertainty*. Quantifying uncertainty is especially critical in real-world settings, which often involve input distributions that are shifted from the training distribution due to a variety of factors including sample bias and non-stationarity. In such settings, well calibrated uncertainty estimates convey information about when a model's output should (or should not) be trusted. Many probabilistic deep learning methods, including Bayesian-and non-Bayesian methods, have been proposed in the literature for quantifying predictive uncertainty, but to our knowledge there has not previously been a rigorous large-scale empirical comparison of these methods under dataset shift. We present a large-scale benchmark of existing state-of-the-art methods on classification problems and investigate the effect of dataset shift on accuracy and calibration. We find that traditional post-hoc calibration does indeed fall short, as do several other previous methods. However, some methods that marginalize over models give surprisingly strong results across a broad spectrum of tasks.

## 1 Introduction

Recent successes across a variety of domains have led to the widespread deployment of deep neural networks (DNNs) in practice. Consequently, the predictive distributions of these models are increasingly being used to make decisions in important applications ranging from machine-learning aided medical diagnoses from imaging (Esteva et al., 2017) to self-driving cars (Bojarski et al., 2016). Such high-stakes applications require not only point predictions but also accurate quantification of predictive uncertainty, i.e. meaningful confidence values in addition to class predictions. With sufficient independent labeled samples from a target data distribution, one can estimate how well

---

[*]Equal contribution
[†]AI Resident
[‡]Corresponding authors

a model's confidence aligns with its accuracy and adjust the predictions accordingly. However, in practice, once a model is deployed the distribution over observed data may shift and eventually be very different from the original training data distribution. Consider, e.g., online services for which the data distribution may change with the time of day, seasonality or popular trends. Indeed, robustness under conditions of distributional shift and out-of-distribution (OOD) inputs is necessary for the safe deployment of machine learning (Amodei et al., 2016). For such settings, calibrated predictive uncertainty is important because it enables accurate assessment of risk, allows practitioners to know how accuracy may degrade, and allows a system to abstain from decisions due to low confidence.

A variety of methods have been developed for quantifying predictive uncertainty in DNNs. Probabilistic neural networks such as mixture density networks (MacKay & Gibbs, 1999) capture the inherent ambiguity in outputs for a given input, also referred to as *aleatoric uncertainty* (Kendall & Gal, 2017). Bayesian neural networks learn a posterior distribution over parameters that quantifies parameter uncertainty, a type of *epistemic uncertainty* that can be reduced through the collection of additional data. Popular approximate Bayesian approaches include Laplace approximation (MacKay, 1992), variational inference (Graves, 2011; Blundell et al., 2015), dropout-based variational inference (Gal & Ghahramani, 2016; Kingma et al., 2015), expectation propagation Hernández-Lobato & Adams (2015) and stochastic gradient MCMC (Welling & Teh, 2011). Non-Bayesian methods include training multiple probabilistic neural networks with bootstrap or ensembling (Osband et al., 2016; Lakshminarayanan et al., 2017). Another popular non-Bayesian approach involves re-calibration of probabilities on a held-out validation set through temperature scaling (Platt, 1999), which was shown by Guo et al. (2017) to lead to well-calibrated predictions on the i.i.d. test set.

**Using Distributional Shift to Evaluate Predictive Uncertainty** While previous work has evaluated the quality of predictive uncertainty on OOD inputs (Lakshminarayanan et al., 2017), there has not to our knowledge been a comprehensive evaluation of uncertainty estimates from different methods under dataset shift. Indeed, we suggest that effective evaluation of predictive uncertainty is most meaningful under conditions of distributional shift. One reason for this is that post-hoc calibration gives good results in independent and identically distributed (i.i.d.) regimes, but can fail under even a mild shift in the input data. And in real world applications, as described above, distributional shift is widely prevalent. Understanding questions of risk, uncertainty, and trust in a model's output becomes increasingly critical as shift from the original training data grows larger.

**Contributions** In the spirit of calls for more rigorous understanding of existing methods (Lipton & Steinhardt, 2018; Sculley et al., 2018; Rahimi & Recht, 2017), this paper provides a benchmark for evaluating uncertainty that focuses not only on the i.i.d. setting but also *uncertainty under distributional shift*. We present a large-scale evaluation of popular approaches in probabilistic deep learning, focusing on methods that operate well in large-scale settings, and evaluate them on a diverse range of classification benchmarks across image, text, and categorical modalities. We use these experiments to evaluate the following questions:

- How trustworthy are the uncertainty estimates of different methods under dataset shift?
- Does calibration in the i.i.d. setting translate to calibration under dataset shift?
- How do uncertainty and accuracy of different methods co-vary under dataset shift? Are there methods that consistently do well in this regime?

In addition to answering the questions above, our code is made available open-source along with our model predictions such that researchers can easily evaluate their approaches on these benchmarks [4].

## 2 Background

**Notation and Problem Setup** Let $x \in \mathbb{R}^d$ represent a set of $d$-dimensional features and $y \in \{1, \ldots, k\}$ denote corresponding labels (targets) for $k$-class classification. We assume that a training dataset $\mathcal{D}$ consists of $N$ i.i.d. samples $\mathcal{D} = \{(x_n, y_n)\}_{n=1}^N$.

Let $p^*(x, y)$ denote the true distribution (unknown, observed only through the samples $\mathcal{D}$), also referred to as the *data generating process*. We focus on classification problems, in which the true distribution is assumed to be a discrete distribution over $k$ classes, and the observed $y \in \{1, \ldots, k\}$

is a sample from the conditional distribution $p^*(y|\boldsymbol{x})$. We use a neural network to model $p_{\boldsymbol{\theta}}(y|\boldsymbol{x})$ and estimate the parameters $\boldsymbol{\theta}$ using the training dataset. At test time, we evaluate the model predictions against a test set, sampled from the same distribution as the training dataset. However, here we also evaluate the model against OOD inputs sampled from $q(\boldsymbol{x}, y) \neq p^*(\boldsymbol{x}, y)$. In particular, we consider two kinds of shifts:

- *shifted versions* of the test inputs where the ground truth label belongs to one of the $k$ classes. We use shifts such as corruptions and perturbations proposed by Hendrycks & Dietterich (2019), and ideally would like the model predictions to become more uncertain with increased shift, assuming shift degrades accuracy. This is also referred to as *covariate shift* (Sugiyama et al., 2017).

- *a completely different OOD dataset*, where the ground truth label is not one of the $k$ classes. Here we check if the model exhibits higher predictive uncertainty for those new instances and to this end report diagnostics that rely only on predictions and not ground truth labels.

**High-level overview of existing methods** A large variety of methods have been developed to either provide higher quality uncertainty estimates or perform OOD detection to inform model confidence. These can roughly be divided into:

1. Methods which deal with $p(y|\boldsymbol{x})$ only, we discuss these in more detail in Section 3.

2. Methods which model the joint distribution $p(y, \boldsymbol{x})$, e.g. deep hybrid models (Kingma et al., 2014; Alemi et al., 2018; Nalisnick et al., 2019; Behrmann et al., 2018).

3. Methods with an OOD-detection component in addition to $p(y|\boldsymbol{x})$ (Bishop, 1994; Lee et al., 2018; Liang et al., 2018), and related work on selective classification (Geifman & El-Yaniv, 2017).

We refer to Shafaei et al. (2018) for a recent summary of these methods. Due to the differences in modeling assumptions, a fair comparison between these different classes of methods is challenging; for instance, some OOD detection methods rely on knowledge of a known OOD set, or train using a none-of-the-above class, and it may not always be meaningful to compare predictions from these methods with those obtained from a Bayesian DNN. We focus on methods described by (1) above, as this allows us to focus on methods which make the same modeling assumptions about data and differ only in how they quantify predictive uncertainty.

## 3   Methods and Metrics

We select a subset of methods from the probabilistic deep learning literature for their prevalence, scalability and practical applicability[5]. These include (see also references within):

- (*Vanilla*) Maximum softmax probability (Hendrycks & Gimpel, 2017)

- (*Temp Scaling*) Post-hoc calibration by temperature scaling using a validation set (Guo et al., 2017)

- (*Dropout*) Monte-Carlo Dropout (Gal & Ghahramani, 2016; Srivastava et al., 2015) with rate $p$

- (*Ensembles*) Ensembles of $M$ networks trained independently on the entire dataset using random initialization (Lakshminarayanan et al., 2017) (we set $M = 10$ in experiments below)

- (*SVI*) Stochastic Variational Bayesian Inference for deep learning (Blundell et al., 2015; Graves, 2011; Louizos & Welling, 2017, 2016; Wen et al., 2018). We refer to Appendix A.6 for details of our SVI implementation.

- (LL) Approx. Bayesian inference for the parameters of the last layer only (Riquelme et al., 2018)

   - (*LL SVI*) Mean field stochastic variational inference on the last layer only
   - (*LL Dropout*) Dropout only on the activations before the last layer

In addition to metrics (we use arrows to indicate which direction is better) that do not depend on predictive uncertainty, such as classification accuracy ↑, the following metrics are commonly used:

**Negative Log-Likelihood (NLL)** ↓ Commonly used to evaluate the quality of model uncertainty on some held out set. *Drawbacks:* Although a proper scoring rule (Gneiting & Raftery, 2007), it can over-emphasize tail probabilities (Quinonero-Candela et al., 2006).

**Brier Score** ↓ (Brier, 1950) Proper scoring rule for measuring the accuracy of predicted probabilities. It is computed as the squared error of a predicted probability *vector*, $p(y|x_n, \boldsymbol{\theta})$, and the one-hot encoded true response, $y_n$. That is,

$$\text{BS} = |\mathcal{Y}|^{-1} \sum_{y \in \mathcal{Y}} (p(y|\boldsymbol{x}_n, \boldsymbol{\theta}) - \delta(y - y_n))^2 = |\mathcal{Y}|^{-1} \Big(1 - 2p(y_n|\boldsymbol{x}_n, \boldsymbol{\theta}) + \sum_{y \in \mathcal{Y}} p(y|\boldsymbol{x}_n, \boldsymbol{\theta})^2\Big). \quad (1)$$

The Brier score has a convenient interpretation as $BS = \text{uncertainty} - \text{resolution} + \text{reliability}$, where uncertainty is the marginal uncertainty over labels, resolution measures the deviation of individual predictions against the marginal, and reliability measures calibration as the average violation of long-term true label frequencies. We refer to DeGroot & Fienberg (1983) for the decomposition of Brier score into calibration and refinement for classification and to (Bröcker, 2009) for the general decomposition for any proper scoring rule. *Drawbacks:* Brier score is insensitive to predicted probabilities associated with in/frequent events.

Both the Brier score and the negative log-likelihood are proper scoring rules and therefore the optimum score corresponds to a perfect prediction. In addition to these two metrics, we also evaluate two metrics—*expected calibration error* and *entropy*. Neither of these is a proper scoring rule, and thus there exist trivial solutions which yield optimal scores; for example, returning the marginal probability $p(y)$ for every instance will yield perfectly calibrated but uninformative predictions. Each proper scoring rule induces a calibration measure (Bröcker, 2009). However, ECE is not the result of such decomposition and has no corresponding proper scoring rule; we instead include ECE because it is popularly used and intuitive. Each proper scoring rule is also associated with a corresponding entropy function and Shannon entropy is that for log probability (Gneiting & Raftery, 2007).

**Expected Calibration Error (ECE)** ↓ Measures the correspondence between predicted probabilities and empirical accuracy (Naeini et al., 2015). It is computed as the average gap between within bucket accuracy and within bucket predicted probability for $S$ buckets $B_s = \{n \in 1 \dots N : p(y_n|\boldsymbol{x}_n, \boldsymbol{\theta}) \in (\rho_s, \rho_{s+1}]\}$. That is, $\text{ECE} = \sum_{s=1}^{S} \frac{|B_s|}{N} |\operatorname{acc}(B_s) - \operatorname{conf}(B_s)|$, where $\operatorname{acc}(B_s) = |B_s|^{-1} \sum_{n \in B_s} [y_n = \hat{y}_n]$, $\operatorname{conf}(B_s) = |B_s|^{-1} \sum_{n \in B_s} p(\hat{y}_n|\boldsymbol{x}_n, \boldsymbol{\theta})$, and $\hat{y}_n = \arg\max_y p(y|\boldsymbol{x}_n, \boldsymbol{\theta})$ is the $n$-th prediction. When bins $\{\rho_s : s \in 1 \dots S\}$ are quantiles of the held-out predicted probabilities, $|B_s| \approx |B_k|$ and the estimation error is approximately constant. *Drawbacks:* Due to binning, ECE does not monotonically increase as predictions approach ground truth. If $|B_s| \neq |B_k|$, the estimation error varies across bins.

There is no ground truth label for fully OOD inputs. Thus we report histograms of **confidence** and predictive **entropy** on known and OOD inputs and **accuracy versus confidence plots** (Lakshminarayanan et al., 2017): Given the prediction $p(y = k|\boldsymbol{x}_n, \boldsymbol{\theta})$, we define the predicted label as $\hat{y}_n = \arg\max_y p(y|\boldsymbol{x}_n, \boldsymbol{\theta})$, and the confidence as $p(y = \hat{y}|\boldsymbol{x}, \boldsymbol{\theta}) = \max_k p(y = k|\boldsymbol{x}_n, \boldsymbol{\theta})$. We filter out test examples corresponding to a particular confidence threshold $\tau \in [0, 1]$ and compute the accuracy on this set.

## 4 Experiments and Results

We evaluate the behavior of the predictive uncertainty of deep learning models on a variety of datasets across three different modalities: images, text and categorical (online ad) data. For each we follow standard training, validation and testing protocols, but we additionally evaluate results on increasingly shifted data and an OOD dataset. We detail the models and implementations used in Appendix A. Hyperparameters were tuned for all methods using Bayesian optimization (Golovin et al., 2017) (except on ImageNet) as detailed in Appendix A.8.

### 4.1 An illustrative example - MNIST

We first illustrate the problem setup and experiments using the MNIST dataset. We used the LeNet (LeCun et al., 1998) architecture, and, as with all our experiments, we follow standard training, validation, testing and hyperparameter tuning protocols. However, we also compute predictions on increasingly shifted data (in this case increasingly rotated or horizontally translated images) and study

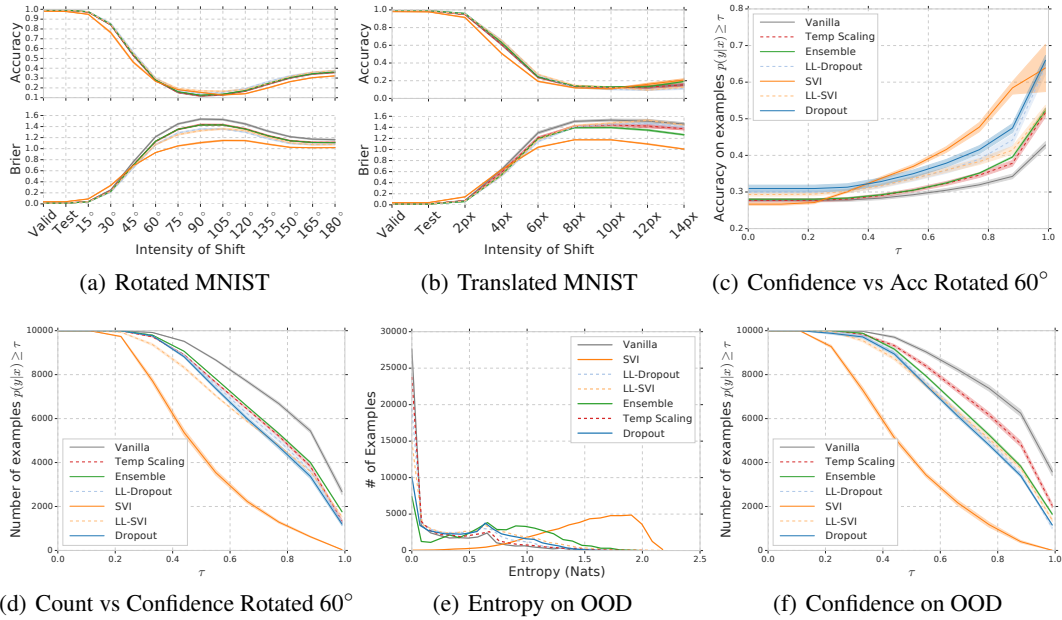

(a) Rotated MNIST      (b) Translated MNIST      (c) Confidence vs Acc Rotated $60°$

(d) Count vs Confidence Rotated $60°$      (e) Entropy on OOD      (f) Confidence on OOD

Figure 1: Results on MNIST: 1(a) and 1(b) show accuracy and Brier score as the data is increasingly shifted. Shaded regions represent standard error over 10 runs. To understand the discrepancy between accuracy and Brier score, we explore the predictive distributions of each method by looking at the confidence of the predictions in 1(c) and 1(d). We also explore the entropy and confidence of each method on entirely OOD data in 1(e) and 1(f). SVI has lower accuracy on the validation and test splits, but it is significantly more robust to dataset shift as evidenced by a lower Brier score, lower overall confidence 1(d) and higher predictive entropy under shift (1(c)) and OOD data (1(e),1(f)).

the behavior of the predictive distributions of the models. In addition, we predict on a completely OOD dataset, Not-MNIST (Bulatov, 2011), and observe the entropy of the model's predictions. We summarize some of our findings in Figure 1 and discuss below.

**What we would like to see:** Naturally, we expect the accuracy of a model to degrade as it predicts on increasingly shifted data, and ideally this reduction in accuracy would coincide with increased forecaster entropy. A model that was well-calibrated on the training and validation distributions would ideally remain so on shifted data. If calibration (ECE or Brier reliability) remained as consistent as possible, practitioners and downstream tasks could take into account that a model is becoming increasingly uncertain. On the completely OOD data, one would expect the predictive distributions to be of high entropy. Essentially, we would like the predictions to indicate that a model "knows what it does not know" due to the inputs straying away from the training data distribution.

**What we observe:** We see in Figures 1(a) and 1(b) that accuracy certainly degrades as a function of shift for all methods tested, and they are difficult to disambiguate on that metric. However, the Brier score paints a clearer picture and we see a significant difference between methods, i.e. prediction quality degrades more significantly for some methods than others. An important observation is that *while calibrating on the validation set leads to well-calibrated predictions on the test set, it does not guarantee calibration on shifted data*. In fact, nearly all other methods (except vanilla) perform better than the state-of-the-art post-hoc calibration (Temperature scaling) in terms of Brier score under shift. While SVI achieves the worst accuracy on the test set, it actually outperforms all other methods by a much larger margin when exposed to significant shift. In Figures 1(c) and 1(d) we look at the distribution of confidences for each method to understand the discrepancy between metrics. We see in Figure 1(d) that SVI has the lowest confidence in general but in Figure 1(c) we observe that SVI gives the highest accuracy at high confidence (or conversely is much less frequently confidently wrong), which can be important for high-stakes applications. Most methods demonstrate very low entropy (Figure 1(e)) and give high confidence predictions (Figure 1(f)) on data that is entirely OOD, i.e. they are confidently wrong about completely OOD data.

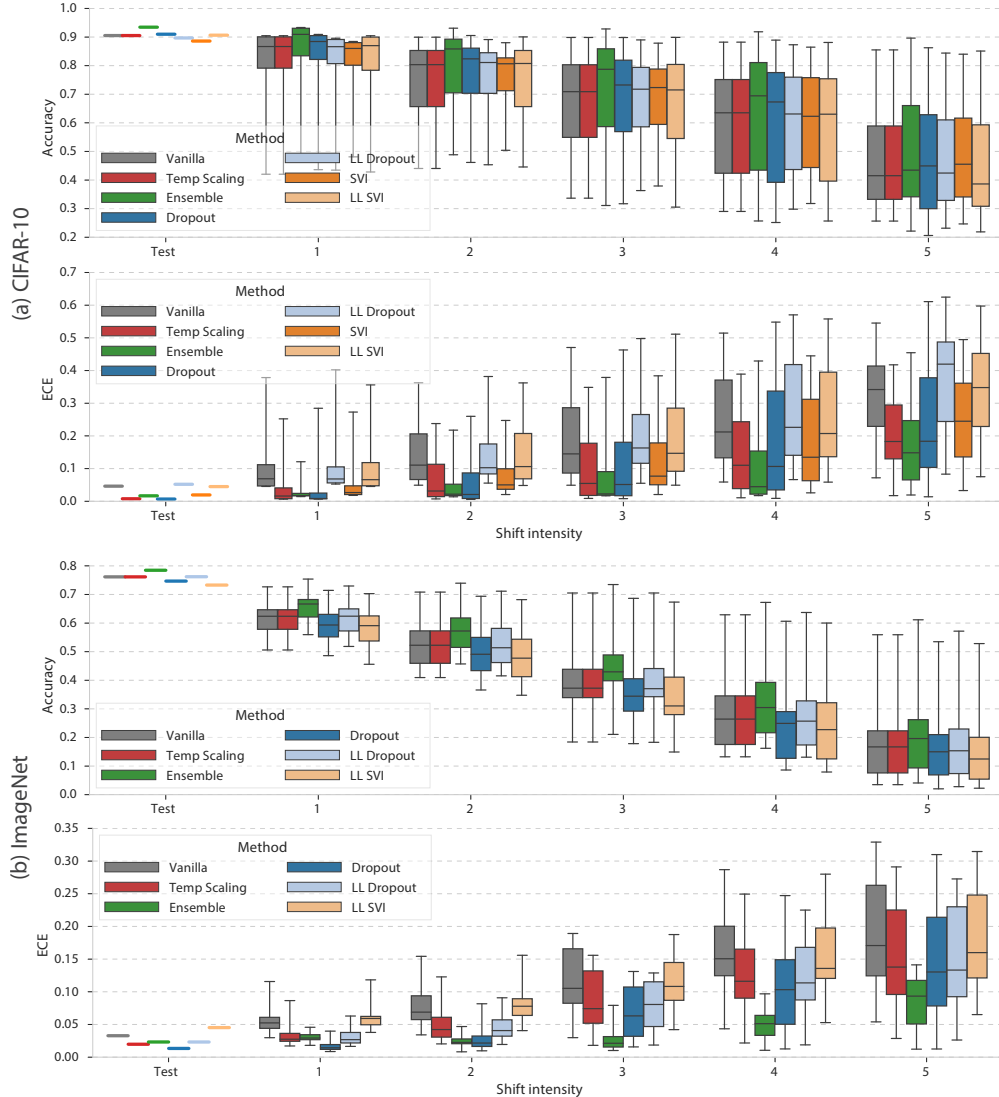

Figure 2: Calibration under distributional shift: a detailed comparison of accuracy and ECE under all types of corruptions on (a) CIFAR-10 and (b) ImageNet. For each method we show the mean on the test set and summarize the results on each intensity of shift with a box plot. Each box shows the quartiles summarizing the results across all (16) types of shift while the error bars indicate the min and max across different shift types. Figures showing additional metrics are provided in Figures S4 (CIFAR-10) and S5 (ImageNet). Tables for numerical comparisons are provided in Appendix G.

## 4.2 Image Models: CIFAR-10 and ImageNet

We now study the predictive distributions of residual networks (He et al., 2016) trained on two benchmark image datasets, CIFAR-10 (Krizhevsky, 2009) and ImageNet (Deng et al., 2009), under distributional shift. We use 20-layer and 50-layer ResNets for CIFAR-10 and ImageNet respectively. For shifted data we use 80 different distortions (16 different types with 5 levels of intensity each, see Appendix B for illustrations) introduced by Hendrycks & Dietterich (2019). To evaluate predictions of CIFAR-10 models on entirely OOD data, we use the SVHN dataset (Netzer et al., 2011).

Figure 2 summarizes the accuracy and ECE for CIFAR-10 (top) and ImageNet (bottom) across all 80 combinations of corruptions and intensities from (Hendrycks & Dietterich, 2019). Figure 3 inspects the predictive distributions of the models on CIFAR-10 (top) and ImageNet (bottom) for shifted (Gaussian blur) and OOD data. Classifiers on both datasets show poorer accuracy and calibration with increasing shift. Comparing accuracy for different methods, we see that ensembles achieve

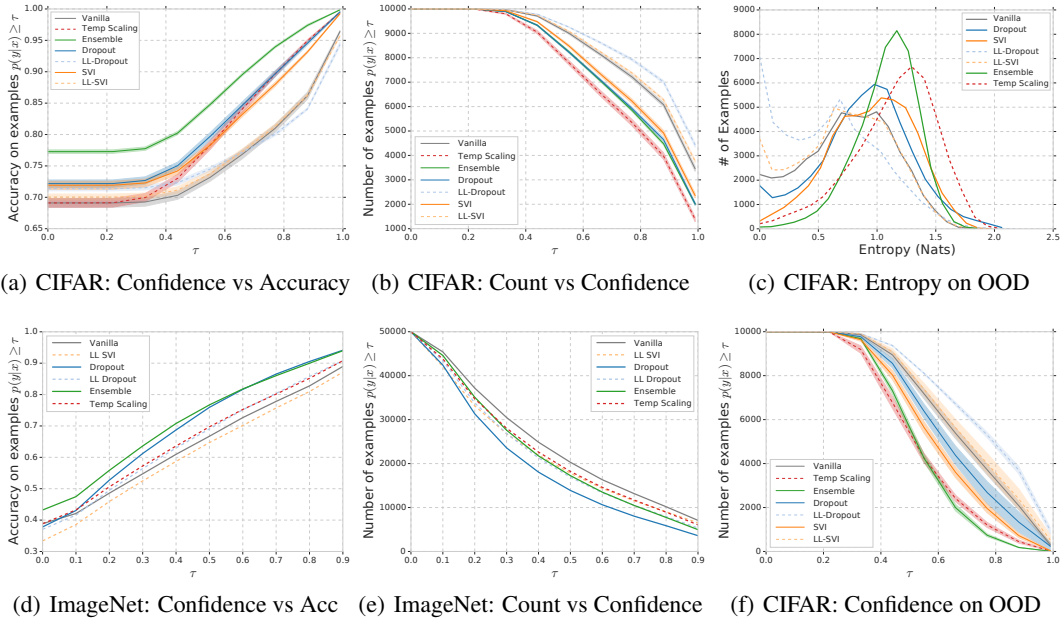

(a) CIFAR: Confidence vs Accuracy    (b) CIFAR: Count vs Confidence    (c) CIFAR: Entropy on OOD

(d) ImageNet: Confidence vs Acc    (e) ImageNet: Count vs Confidence    (f) CIFAR: Confidence on OOD

Figure 3: Results on CIFAR-10 and ImageNet. Left column: 3(a) and 3(d) show accuracy as a function of confidence. Middle column: 3(b) and 3(e) show the number of examples greater than given confidence values for Gaussian blur of intensity 3. Right column: 3(c) and 3(f) show histogram of entropy and confidences from CIFAR-trained models on a completely different dataset (SVHN).

highest accuracy under distributional shift. Comparing the ECE for different methods, we observe that while the methods achieve comparable low values of ECE for small values of shift, ensembles outperform the other methods for larger values of shift. To test whether this result is due simply to the larger aggregate capacity of the ensemble, we trained models with double the number of filters for the Vanilla and Dropout methods. The higher-capacity models showed no better accuracy or calibration for medium- to high-shift than the corresponding lower-capacity models (see Appendix C). In Figures S8 and S9 we also explore the effect of the number of samples used in dropout, SVI and last layer methods and size of the ensemble, on CIFAR-10. We found that while increasing ensemble size up to 50 did help, most of the gains of ensembling could be achieved with only 5 models. Interestingly, *while temperature scaling achieves low ECE for low values of shift, the ECE increases significantly as the shift increases, which indicates that calibration on the i.i.d. validation dataset does not guarantee calibration under distributional shift*. (Note that for ImageNet, we found similar trends considering just the top-5 predicted classes, See Figure S5.) Furthermore, the results show that while temperature scaling helps significantly over the vanilla method, ensembles and dropout tend to be better. In Figure 3, we see that ensembles and dropout are more accurate at higher confidence. However, in 3(c) we see that temperature scaling gives the highest entropy on OOD data. Ensembles consistently have high accuracy but also high entropy on OOD data. We refer to Appendix C for additional results; Figures S4 and S5 report additional metrics on CIFAR-10 and ImageNet, such as Brier score (and its component terms), as well as top-5 error for increasing values of shift.

Overall, ensembles consistently perform best across metrics and dropout consistently performed better than temperature scaling and last layer methods. *While the relative ordering of methods is consistent on both CIFAR-10 and ImageNet (ensembles perform best), the ordering is quite different from that on MNIST where SVI performs best.* Interestingly, LL-SVI and LL-Dropout perform worse than the vanilla method on shifted datasets as well as SVHN. We also evaluate a variational Gaussian process as a last layer method in Appendix E but it did not outperform LL-SVI and LL-Dropout.

### 4.3 Text Models

Following Hendrycks & Gimpel (2017), we train an LSTM (Hochreiter & Schmidhuber, 1997) on the 20newsgroups dataset (Lang, 1995) and assess the model's robustness under distributional shift

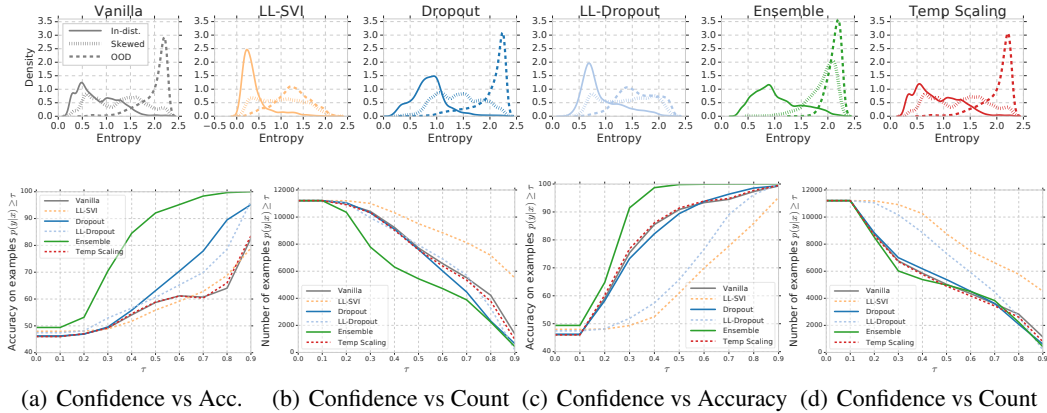

(a) Confidence vs Acc.  (b) Confidence vs Count  (c) Confidence vs Accuracy  (d) Confidence vs Count

Figure 4: Top row: Histograms of the entropy of the predictive distributions for in-distribution (solid lines), shifted (dotted lines), and completely different OOD (dashed lines) text examples. Bottom row: Confidence score vs accuracy and count respectively when evaluated for in-distribution and in-distribution shift text examples (a,b), and in-distribution and OOD text examples (c,d).

and OOD text. We use the even-numbered classes (10 classes out of 20) as in-distribution and the 10 odd-numbered classes as shifted data. We provide additional details in Appendix A.4.

We look at confidence vs accuracy when the test data consists of a mix of in-distribution and either shifted or completely OOD data, in this case the One Billion Word Benchmark (LM1B) (Chelba et al., 2013). Figure 4 (bottom row) shows the results. Ensembles significantly outperform all other methods, and achieve better trade-off between accuracy versus confidence. Surprisingly, LL-Dropout and LL-SVI perform worse than the vanilla method, giving higher confidence incorrect predictions, especially when tested on fully OOD data.

Figure 4 reports histograms of predictive entropy on in-distribution data and compares them to those for the shifted and OOD datasets. This reflects how amenable each method is to abstaining from prediction by applying a threshold on the entropy. As expected, most methods achieve the highest predictive entropy on the completely OOD dataset, followed by the shifted dataset and then the in-distribution test dataset. Only ensembles have consistently higher entropy on the shifted data, which explains why they perform best on the confidence vs accuracy curves in the second row of Figure 4. Compared with the vanilla model, Dropout and LL-SVI have more a distinct separation between in-distribution and shifted or OOD data. While Dropout and LL-Dropout perform similarly on in-distribution, LL-Dropout exhibits less uncertainty than Dropout on shifted and OOD data. Temperature scaling does not appear to increase uncertainty significantly on the shifted data.

## 4.4 Ad-Click Model with Categorical Features

Finally, we evaluate the performance of different methods on the *Criteo Display Advertising Challenge*[6] dataset, a binary classification task consisting of 37M examples with 13 numerical and 26 categorical features per example. We introduce shift by reassigning each categorical feature to a random new token with some fixed probability that controls the intensity of shift. This coarsely simulates a type of shift observed in non-stationary categorical features as category tokens appear and disappear over time, for example due to hash collisions. The model consists of a 3-hidden-layer multi-layer-perceptron (MLP) with hashed and embedded categorical features and achieves a negative log-likelihood of approximately 0.5 (contest winners achieved 0.44). Due to class imbalance ($\sim 25\%$ of examples are positive), we report AUC instead of classification accuracy.

Results from these experiments are depicted in Figure 5. (Figure S7 in Appendix C shows additional results including ECE and Brier score decomposition.) We observe that ensembles are superior in terms of both AUC and Brier score for most of the values of shift, with the performance gap between ensembles and other methods generally increasing as the shift increases. Both Dropout model variants yielded improved AUC on shifted data, and Dropout surpassed ensembles in Brier

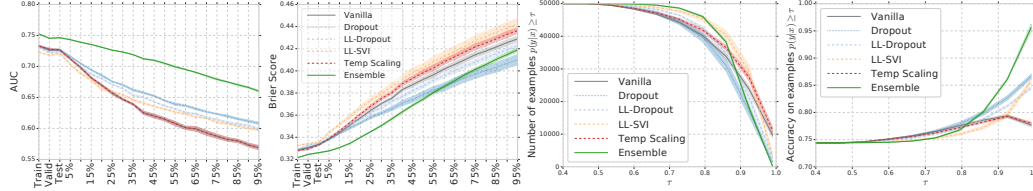

Figure 5: Results on Criteo: The first two plots show degrading AUCs and Brier scores with increasing shift while the latter two depict the distribution of prediction confidences and their corresponding accuracies at 75% randomization of categorical features. SVI is excluded as it performed too poorly.

score at shift-randomization values above 60%. SVI proved challenging to train, and the resulting model uniformly performed poorly; LL-SVI fared better but generally did not improve upon the vanilla model. *Strikingly, temperature scaling has a worse Brier score than Vanilla indicating that post-hoc calibration on the validation set actually harms calibration under dataset shift.*

## 5 Takeaways and Recommendations

We presented a large-scale evaluation of different methods for quantifying predictive uncertainty under dataset shift, across different data modalities and architectures. Our take-home messages are the following:

- Along with accuracy, the quality of uncertainty consistently degrades with increasing dataset shift regardless of method.

- Better calibration and accuracy on the i.i.d. test dataset does not usually translate to better calibration under dataset shift (shifted versions as well as completely different OOD data).

- Post-hoc calibration (on i.i.d validation) with temperature scaling leads to well-calibrated uncertainty on the i.i.d. test set and small values of shift, but is significantly outperformed by methods that take epistemic uncertainty into account as the shift increases.

- Last layer Dropout exhibits less uncertainty on shifted and OOD datasets than Dropout.

- SVI is very promising on MNIST/CIFAR but it is difficult to get to work on larger datasets such as ImageNet and other architectures such as LSTMs.

- The relative ordering of methods is mostly consistent (except for MNIST) across our experiments. The relative ordering of methods on MNIST is not reflective of their ordering on other datasets.

- Deep ensembles seem to perform the best across most metrics and be more robust to dataset shift. We found that relatively small ensemble size (e.g. $M = 5$) may be sufficient (Appendix D).

- We also compared the set of methods on a real-world challenging genomics problem from Ren et al. (2019). Our observations were consistent with the other experiments in the paper. Deep ensembles performed best, but there remains significant room for improvement, as with the other experiments in the paper. See Section F for details.

We hope that this benchmark is useful to the community and inspires more research on uncertainty under dataset shift, which seems challenging for existing methods. While we focused only on the quality of predictive uncertainty, applications may also need to consider computational and memory costs of the methods; Table S1 in Appendix A.9 discusses these costs, and the best performing methods tend to be more expensive. Reducing the computational and memory costs, while retaining the same performance under dataset shift, would also be a key research challenge.

**Acknowledgements**

We thank Alexander D'Amour, Jakub Świątkowski and our reviewers for helpful feedback that improved the manuscript.

## Footnotes

[4]https://github.com/google-research/google-research/tree/master/uq_benchmark_2019

[5]The methods used scale well for training and prediction (see **in Appendix A.9.**). We also explored methods such as scalable extensions of Gaussian Processes (Hensman et al., 2015), but they were challenging to train on the 37M example Criteo dataset or the 1000 classes of ImageNet.

[6]https://www.kaggle.com/c/criteo-display-ad-challenge

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
