[Supplementary Material]

# Can You Trust Your Model's Uncertainty? Evaluating Predictive Uncertainty Under Dataset Shift: Appendix

## A    Model Details

### A.1    MNIST

We evaluated both LeNet and a fully-connected neural network (MLP) under shift on MNIST. We observed similar trends across metrics for both models, so we report results only for LeNet in Section 4.1. LeNet and MLP were trained for 20 epochs using the Adam optimizer (Kingma & Ba, 2014) and used ReLU activation functions. For stochastic methods, we averaged 300 sample predictions to yield a predictive distribution, and the ensemble model used 10 instances trained from independent random initializations. The MLP architecture consists of two hidden layers of 200 units each with dropout applied before every dense layer. The LeNet architecture (LeCun et al., 1998) applies two convolutional layers 3x3 kernels of 32 and 64 filters respectively) followed by two fully-connected layers with one hidden layer of 128 activations; dropout was applied before each fully-connected layer. We employed hyperparameter tuning (See Section A.8) to select the training batch size, learning rate, and dropout rate.

### A.2    CIFAR-10

Our CIFAR model used the ResNet-20 V1 architecture with ReLU activations. Model parameters were trained for 200 epochs using the Adam optimizer and employed a learning rate schedule that multiplied an initial learning rate by 0.1, 0.01, 0.001, and 0.0005 at steps 80, 120, 160, and 180 respectively. Training inputs were randomly distorted using horizontal flips and random crops preceded by 4-pixel padding as described in (He et al., 2016). For relevant methods, dropout was applied before each convolutional and dense layer (excluding the raw inputs), and stochastic methods sampled 128 predictions per sample. Hyperparameter tuning was used to select the initial learning rate, training batch size, and the dropout rate.

### A.3    ImageNet 2012

Our ImageNet model used the ResNet-50 V1 architecture with ReLU activations and was trained for 90 epochs using SGD with Nesterov momentum. The learning rate schedule linearly ramps up to a base rate in 5 epochs and scales down by a factor of 10 at each of epochs 30, 60, and 80. As with the CIFAR-10 model, stochastic methods used a sample-size of 128. Training images were distorted with random horizontal flips and random crops.

### A.4    20 Newsgroups

We use a pre-processing strategy similar to the one proposed by Hendrycks & Gimpel (2017) for 20 Newsgroups. We build a vocabulary of size 30,000 words and words are indexed based on the word frequencies. The rare words are encoded as unknown words. We fix the length of each text input by setting a limit of 250 words, and those longer than 250 words are truncated, and those shorter than 250 words are padded with zeros. Text in even-numbered classes are used as in-distribution inputs, and text from the odd-numbered of classes are used shifted OOD inputs. A dataset with the same number of randomly selected text inputs from the LM1B dataset (Chelba et al., 2013) is used as completely different OOD dataset. The classifier is trained and evaluated only using the text from the even-numbered in-distribution classes in the training dataset. The final test results are evaluated based on in-distribution test dataset, shift OOD test dataset, and LM1B dataset.

The vanilla model uses a one-layer LSTM model of size 32 and a dense layer to predict the 10 class probabilities based on word embedding of size 128. A dropout rate of 0.1 is applied to both the LSTM layer and the dense layer for the Dropout model. The LL-SVI model replaces the last dense layer with a Bayesian layer, the ensemble model aggregates 10 vanilla models, and stochastic methods sample 5 predictions per example. The vanilla model accuracy for in-distribution test data is 0.955.

## A.5 Criteo

Each categorical feature $x_k$ from the Criteo dataset was encoded by hashing the string token into a fixed number of buckets $N_k$ and either encoding the hash-bin as a one-hot vector if $N_k < 110$ or embedding each bucket as a $d_k$ dimensional vector otherwise. This dense feature vector, concatenated with 13 numerical features, feeds into a batch-norm layer followed by a 3-hidden-layer MLP. Each model was trained for one epoch using the Adam optimizer with a non-decaying learning rate.

Values of $N_k$ and $d_k$ were tuned to maximize log-likelihood for a vanilla model, and the resulting architectural parameters were applied to all methods. This tuning yielded hidden-layers of size 2572, 1454, and 1596, and hash-bucket counts and embedding dimensions of sizes listed below:

$$N_k = [1373, 2148, 4847, 9781, 396, 28, 3591, 2798, 14, 7403, 2511, 5598, 9501,$$
$$46, 4753, 4056, 23, 3828, 5856, 12, 4226, 23, 61, 3098, 494, 5087]$$
$$d_k = [3, 9, 29, 11, 17, 0, 14, 4, 0, 12, 19, 24, 29, 0, 13, 25, 0, 8, 29, 0, 22, 0, 0, 31, 0, 29]$$

Learning rate, batch size, and dropout rate were further tuned for each method. Stochastic methods used 128 prediction samples per example.

## A.6 Stochastic Variational Inference Details

For MNIST we used Flipout (Wen et al., 2018), where we replaced each dense layer and convolutional layer with mean-field variational dense and convolutional Flipout layers respectively. Variational inference for deep ResNets (He et al., 2016) is non-trivial, so for CIFAR we replaced a single linear layer per residual branch with a Flipout layer, removed batch normalization, added Selu non-linearities (Klambauer et al., 2017), empirical Bayes for the prior standard deviations as in Wu et al. (2019) and careful tuning of the initialization via Bayesian optimization.

## A.7 Variational Gaussian Process Details

For the experiments where Gaussian Processes were compared, we used Variational Gaussian Processes to fit the model logits as in Hensman et al. (2015). These were then passed through a Categorical distribution and numerically integrated over using Gauss-Hermite quadrature. Each class was treated as a separate Gaussian Process, with 100 inducing points used for each class. The inducing points were initialized with model outputs on random dataset examples for CIFAR, and with Gaussian noise for MNIST. Uniform noise inducing point initialization was also tested but there was negligible difference between the three methods. All zero inducing points initializations numerically failed early on in training. Exponentiated quadratic plus linear kernels were used for all experiments. 250 samples were drawn from the logit distribution during training time to get a better estimate of the ELBO to backpropagate through. 250 logit samples were drawn at test time. $10^{-5} * I$ was added to the diagonal of the covariance matrix to ensure positive definiteness.

We used 100 trials of random hyperparamter settings, selecting the configuration with the best final validation accuracy. The learning rate was tuned in $[10^{-4}, 1.0]$ on a log scale; the initial kernel amplitude in $[-2.0, 2.0]$; the initial kernel length scale in $[-2.0, 2.0]$; the variational distribution covariance was initialized to $s * I$ where $s$ was tuned in $[0.1, 2.0]$; $1 - \beta_1$ in Adam was tuned on $[10^{-2}, 0.15]$ on a log scale.

The Adam optimizer with a batch size of 512 was used, training for the same number of epochs as other methods. The same learning rate schedule was as other methods for the model and kernel parameters, but the learning rate for the variational parameters also included a 5 epoch warmup in order to help with numerical stability.

## A.8 Hyperparameter Tuning

Hyperparameters were optimized through Bayesian optimization using Google Vizier (Golovin et al., 2017). We maximized the log-likelihood on a validation set that was held out from training (10K examples for MNIST and CIFAR-10, 125K examples for ImageNet). We optimized log-likelihood rather than accuracy since the former is a proper scoring rule.

## A.9 Computational and Memory Complexity of Different methods

In addition to performance, applications may also need to consider computational and memory costs; Table S1 discusses them for each method.

Table S1: Computational and memory costs for evaluated methods. Notation: $m$ represents flops or storage for the full model, $d$ represents flops or storage for the last layer, $k$ denotes replications, $z$ the number of inducing points for Gaussian Processes, $n$ denotes number of evaluated points, and $v$ denotes the validation set size. Serving/training compute is identical except that $v = 0$ for serving. Implicit in this table is a memory/compute tradeoff for sampling. Sampled weights/masks need not be stored explicitly via PRNG seed reuse; we assume the computational cost of sampling is zero.

| Method | Compute/$n$ | Storage |
|---|---|---|
| Vanilla | $m$ | $m$ |
| Temp Scaling | $m + vm/n$ | $m$ |
| LL-Dropout | $m + d(k-1)$ | $m$ |
| LL-SVI | $m + d(k-1)$ | $m + d$ |
| SVI | $mk$ | $2m$ |
| Dropout | $mk$ | $m$ |
| Gaussian Process | $m + z^3$ | $m + z^2$ |
| Ensemble | $mk$ | $mk$ |

# B   Shifted Images

We distorted MNIST images using rotations with spline filter interpolation and cyclic translations as depicted in Figure S1.

For the corrupted ImageNet dataset, we used ImageNet-C (Hendrycks & Dietterich, 2019). Figure S2 shows examples of ImageNet-C images at varying corruption intensities. Figure S3 shows ImageNet-C images with the 16 corruptions analyzed in this paper, at intensity 3 (on a scale of 1 to 5).

(a) Rotations

(b) Cyclic translations

Figure S1: Examples of rotated and cyclically translated MNIST digits. Results for accuracy and calibration on rotated/translated MNIST are shown in Figure 1.

Figure S2: Examples of ImageNet images corrupted by Gaussian blur, at intensities of 0 (uncorrupted image) through 5 (maximum corruption included in ImageNet-C).

Figure S3: Examples of 16 corruption types in ImageNet-C images, at corruption intensity 3 (on a scale from 1–5). The same corruptions were applied to CIFAR-10. Figure 2 and Section C show boxplots for each uncertainty method and corruption intensity, spanning all corruption types.

# C  Evaluating uncertainty under distributional shift: Additional Results

Figures S4, S5 and S7 show comprehensive results on CIFAR-10, ImageNet and Criteo respectively across various metrics including Brier score, along with the components of the Brier score : reliability (lower means better calibration) and resolution (higher values indicate better predictive quality). Ensembles and dropout outperform all other methods across corruptions, while LL SVI shows no improvement over the baseline model. Figure S6 shows accuracy and ECE for models with double the number of ResNet filters; the higher-capacity models are not better calibrated than their lower-capacity counterparts, suggesting that the good calibration performance of ensembles is not due simply to higher capacity.

Figure S4: Boxplots facilitating comparison of methods for each shift level showing detailed comparisons of various metrics under all types of corruptions on CIFAR-10. Each box shows the quartiles summarizing the results across all types of shift while the error bars indicate the min and max across different shift types.

Figure S5: Boxplots facilitating comparison of methods for each shift level showing detailed comparisons of various metrics under all types of corruptions on ImageNet. Each box shows the quartiles summarizing the results across all types of shift while the error bars indicate the min and max across different shift types.

Figure S6: Boxplots facilitating comparison of results for higher-capacity models ('Wide Vanilla' and 'Wide Dropout') with their lower-capacity counterparts on CIFAR. Each box shows the quartiles summarizing the results across all types of shift while the error bars indicate the min and max across different shift types.

Figure S7: Comprehensive comparison of metrics on Criteo models. The Brier decomposition reveals that the majority of its degradation is due to worsening reliability, and this component alone appears to largely explain the ranking of methods in total Brier score. Ensemble notably degrades most rapidly in resolution but persists with better reliability compared other methods for most of the data-corruption range; on ECE it remains roughly in the middle among explored methods. Dropout (and to a lesser extend LL-Dropout) perform best on ECE and experience slower degradation in both resolution and reliability leading it to surpass ensembles at the severe range of data corruption. Total Brier score and AUC results are discussed in detail in Section 4.4.

## D   Effect of the number of samples on the quality of uncertainty

Figure S8 shows the effect of the number of sample sizes used by Dropout, SVI (and last-layer variants) on the quality of predictive uncertainty, as measured by the Brier score. Increasing the number of samples has little effect on last-layer variants, whereas increasing the number of samples improves the performance for SVI and Dropout, with diminishing returns beyond size 5.

(a) Dropout

(b) LL-Dropout

(c) SVI

(d) LL-SVI

Figure S8: Effect of Dropout and SVI sample sizes on CIFAR-10 Brier scores under increasing Gaussian blur. See Section 4.2 for full results on CIFAR-10.

Figure S9 shows the effect of ensemble size on CIFAR-10 (top) and ImageNet (bottom). Similar to SVI and Dropout, we see that increasing the number of models in the ensemble improves performance with diminishing returns beyond size 5. As mentioned earlier, the Brier score can be further

decomposed into BS = calibration + refinement = reliability + uncertainty − resolution where reliability ↓ measures calibration as the average violation of long-term true label frequencies, and refinement = uncertainty − resolution, where uncertainty is the marginal uncertainty over labels (independent of predictions) and resolution ↑ measures the deviation of individual predictions from the marginal.

(a) Brier Score　　　　　　　(b) Brier Reliability　　　　　　(c) Brier Resolution

(d) Brier Score　　　　　　　(e) Brier Reliability　　　　　　(f) Brier Resolution

Figure S9: Effect of the ensemble size on CIFAR-10 (top row) and ImageNet (bottom row) Brier scores under increasing Gaussian-blur shift. We additionally show the Brier score components: Reliability (lower means better calibration) and Resolution (higher values indicate better predictive quality). Note that the scales for Reliability are significantly smaller than the other plots.

# E  Variational Gaussian Process Results

(a) Brier Score

(b) Accuracy

(c) ECE

Figure S10: Uncertainty metrics across shift levels on CIFAR-10, where level 0 is the test set, using a last layer Variational Gaussian Process. See **Appendix A.7** for experiment details.

# F  OOD detection for genomic sequences

We studied the set of methods for detecting OOD genomic sequence, as a challenging realistic problem for OOD detection proposed by Ren et al. (2019). Classifiers are trained on 10 in-distribution bacteria classes, and tested for OOD detection of 60 OOD bacteria classes. The model architecture is the same as that in Ren et al. (2019): a convolutional neural networks with 1000 filters of length 20, followed by a global max pooling layer, a dense layer of 1000 units, and a last dense layer that outputs class prediction logits. For the dropout method, we add a dropout layer each after the max pooling layer and the dense layer respectively. For the LL-Dropout method, only a dropout layer after the dense layer is added. We use the dropout rate of 0.2. For the LL-SVI method, we replace the last dense layer with a stochastic variational inference dense layer. The classification accuracy for in-distribution is around 0.8 for the various types of classifiers.

Figure S11 shows the confidence vs (a) accuracy and (b) count when the test data consists of a mix of in-distribution and OOD data. Ensembles significantly outperform all other methods, and achieve better trade-off between accuracy versus confidence. Dropout performs better than Temp Scaling, and they both perform better than LL-Dropout, LL-SVI, and the Vanilla method. Note that the accuracy on examples $p(y|x) \geq 0.9$ for the best method is still below 65%, suggesting that this realistic genomic sequences dataset is a challenging problem to benchmark future methods.

(a) Confidence vs Accuracy          (b) Confidence vs Count

Figure S11: Confidence score vs accuracy and count respectively when evaluated for in-distribution and OOD genomic sequences.

# G  Tables of Metrics

The tables below report quartiles of Brier score, negative log-likelihood, and ECE for each model and dataset where quartiles are computed over all corrupted variants of the dataset.

## G.1  CIFAR-10

| Dataset | Vanilla | Temp. Scaling | Ensembles | Dropout | LL-Dropout | SVI | LL-SVI |
|---|---|---|---|---|---|---|---|
| Brier Score (25th) | 0.243 | 0.227 | 0.165 | 0.215 | 0.259 | 0.250 | 0.246 |
| Brier Score (50th) | 0.425 | 0.392 | 0.299 | 0.349 | 0.416 | 0.363 | 0.431 |
| Brier Score (75th) | 0.747 | 0.670 | 0.572 | 0.633 | 0.728 | 0.604 | 0.732 |
| NLL (25th) | 2.356 | 1.685 | 1.543 | 1.684 | 2.275 | 1.628 | 2.352 |
| NLL (50th) | 1.120 | 0.871 | 0.653 | 0.771 | 1.086 | 0.823 | 1.158 |
| NLL (75th) | 0.578 | 0.473 | 0.342 | 0.446 | 0.626 | 0.533 | 0.591 |
| ECE (25th) | 0.057 | 0.022 | 0.031 | 0.021 | 0.069 | 0.029 | 0.058 |
| ECE (50th) | 0.127 | 0.049 | 0.037 | 0.034 | 0.136 | 0.064 | 0.135 |
| ECE (75th) | 0.288 | 0.180 | 0.110 | 0.174 | 0.292 | 0.187 | 0.275 |

### G.2 ImageNet

| Dataset | Vanilla | Temp. Scaling | Ensembles | Dropout | LL-Dropout | LL-SVI |
|---|---|---|---|---|---|---|
| Brier Score (25th) | 0.553 | 0.551 | 0.503 | 0.577 | 0.550 | 0.590 |
| Brier Score (50th) | 0.733 | 0.726 | 0.667 | 0.754 | 0.723 | 0.766 |
| Brier Score (75th) | 0.914 | 0.899 | 0.835 | 0.922 | 0.896 | 0.938 |
| NLL (25th) | 1.859 | 1.848 | 1.621 | 1.957 | 1.830 | 2.218 |
| NLL (50th) | 2.912 | 2.837 | 2.446 | 3.046 | 2.858 | 3.504 |
| NLL (75th) | 4.305 | 4.186 | 3.661 | 4.567 | 4.208 | 5.199 |
| ECE (25th) | 0.057 | 0.031 | 0.022 | 0.017 | 0.034 | 0.065 |
| ECE (50th) | 0.102 | 0.072 | 0.032 | 0.043 | 0.071 | 0.106 |
| ECE (75th) | 0.164 | 0.129 | 0.053 | 0.109 | 0.123 | 0.148 |

### G.3 Criteo

| Dataset | Vanilla | Temp. Scaling | Ensembles | Dropout | LL-Dropout | SVI | LL-SVI |
|---|---|---|---|---|---|---|---|
| Brier Score (25th) | 0.353 | 0.355 | 0.336 | 0.350 | 0.353 | 0.512 | 0.361 |
| Brier Score (50th) | 0.385 | 0.391 | 0.366 | 0.373 | 0.379 | 0.512 | 0.396 |
| Brier Score (75th) | 0.409 | 0.416 | 0.395 | 0.393 | 0.403 | 0.512 | 0.421 |
| NLL (25th) | 0.581 | 0.594 | 0.508 | 0.532 | 0.542 | 7.479 | 0.554 |
| NLL (50th) | 0.788 | 0.829 | 0.552 | 0.577 | 0.600 | 7.479 | 0.633 |
| NLL (75th) | 0.986 | 1.047 | 0.608 | 0.624 | 0.664 | 7.479 | 0.711 |
| ECE (25th) | 0.041 | 0.055 | 0.044 | 0.043 | 0.052 | 0.254 | 0.066 |
| ECE (50th) | 0.097 | 0.113 | 0.100 | 0.085 | 0.100 | 0.254 | 0.127 |
| ECE (75th) | 0.135 | 0.149 | 0.141 | 0.116 | 0.136 | 0.254 | 0.162 |