[Reviews · NeurIPS 2019]

Reviewer 1



Originality: Many recent attempts and benchmarks have been proposed that look into uncertainty for deep learning models. However, the authors provide a sufficiently diverse suite of tasks, and not just a single task, strengthening their argument, due to transferability to more than one domain. Moreover, to the best of my knowledge, it is the first time that I see a large-scale empirical evaluation of uncertainty calibration methods (e.g. temperature scaling) to Bayesian deep learning methods (e.g. SVI and MC Dropout). Quality & Clarity: The ideas are clearly communicated and the paper is carefully written. The experimental setup is described adequately and the results seem to align with our intuition. Significance: The empirical results and motivation of this paper can be very impactful, especially for practitioners but also for researchers! In case of exact Bayesian inference methods (e.g. HMC), we wouldn’t expect to observe a degradation of uncertainty as to the data distribution shifts, but due to the approximations in the variational methods (e.g. SVI, Dropout) this side-effect emerges! As a result, this paper highlights this and goes against the intuition many may had before deep neural network based models.

Reviewer 2



UPDATE: I thank the authors for their feedback, which I have read. The authors have done well in addressing my (and other) concerns and I am raising my score to an 8. ---------------------------------------- Thanks to the authors for the hard work on this paper. ==Originality== The work is original. I know of no other suites of benchmarks that evaluate predictive uncertainty for a variety of methods on a set of datasets. ==Quality== The work is good. The experiments are well-designed, the metrics are appropriate and informative. There are some questions I had about the work that I wish were answered, and these are listed below. Why does SVI perform well on MNIST but poorly on every other dataset considered? I believe it is within the scope of this work to offer at least a basic explanation of this observation. It would be nice to have a proposed procedure for an applied ML practitioner who wants to compare a variety of uncertainty estimation procedures. Should one create one's own skew and OOD datasets? Are there any principles that are important to keep in mind while doing that? How hard will it be to use your eventually released code to do that? I ask these questions because your paper is extremely useful already, and it would be great to have this additional bridge discussion to allow someone to actually figure out whether the SVI MNIST results are a one-time fluke or if SVI might be great for their own dataset. As of now, the reader has no idea what it means that SVI works well on MNIST but is hard to train and/or poorly performing on the other datasets in this work. Some questions/comments regarding the Ensemble method: -Comparing accuracy between the ensemble method and the other methods is unfair. Ensembles are likely to do better. It might be obvious to some readers, but I recommend pointing this out as well. It does raise the question of whether ensemble method performs best because it has k times the number of parameters as (most of) the other methods. Can this be confirmed somehow? Does more capacity mean better OOD calibration? -In the original Ensembles paper (Lakshminarayanan, 2017), there are two more tricks used to make this method work: (a) the heteroscedastic loss in eq 1 and (b) adversarial training in sec 2.3. Are either of these used in your implementation? Or is it just a plain ensemble? I know the code will be released, and the readers will be able to answer this question themselves, but it's probably worthwhile to address this question in the main text or the appendix. Minor: In section 4.4 SVI is not used, and the reason is explained. In section 4.3 SVI is not used, but the reason is not explained. ==Clarity== The text is clear and well-written. ==Significance== The work is fairly significant. It will ground the study of uncertainty estimation and hopefully provide a standard suite for future researchers to reach for when they are developing new uncertainty estimation algorithms.

Reviewer 3



Strengths – Addresses an important problem, and the need for predictive uncertainty under dataset shift is well-motivated. – Performs and presents a comprehensive set of experiments on several state of the art methods, on datasets across multiple modalities. – Surveys and summarizes the dominant threads in prior work clearly. – Includes a comprehensive supplementary material and experimental details to enable reproducibility. Weaknesses / Questions – The paper conducts a large scale empirical study, and draws direct conclusions. However, there is little by way of analysis / actionable takeaways from these conclusions. For example, have the authors analyzed why SVI / ensembles tend to perform best for MNIST / CIFAR-10 respectively, towards the goal of building a better general understanding of what kinds of approaches are best-suited to any given dataset? – Have the authors investigated the performance on different OOD datasets that are more / less similar to the source dataset (say MSCOCO vs SVHN as OOD for ImageNet), and seen if consistent orderings of the various approaches are observed? – As an analysis paper with several different sets of experiments, I found the experimental section somewhat disorganized. In particular, Figures 1-2, 4 required repeated cross-referencing, and a consistent ordering scheme for plots would have made for a significantly better read. – L180: "SVI .. outperforms all methods by a large margin .. ": While SVI does seem to have lower Brier scores with shift, accuracies don’t appear to be any better (figure 1b) – how was this observation made? – L135: "However, both measure .. not directly measured by proper scoring rules": What are these additional properties that ECE and entropy capture? ==================== Final recommendation ========================= I have read and am satisfied with the author response and will raise my score to 7. I also recommend that (as mentioned in the author response) experiments to better understand the relation of SVI performance and model specification, and better tuning of dropout models be included in the final version.

[Author Response · NeurIPS 2019]

We thank the reviewers for their thoughtful reviews and feedback which will help us improve the paper. Overall, we found the reviews to be very positively worded, noting e.g. that the work is "comprehensive", "thorough", "impactful", "well-motivated", "clearly written", "addresses an important problem", etc. We thank the reviewers for their kind words and will address their comments below. We address common questions first and then address reviewers individually.

**Code release:** We would first like to share that our code and all model predictions from our experiments have been shared publicly online and are already being used by other research groups (for anonymity we will not share links here). We are also continuously refining this benchmark and have added a variational GP classifier as a last layer method.

*"Why does SVI perform well on MNIST but poorly on every other dataset considered?"* (**R2,R3**): We agree that this should be discussed in the paper. Essentially, due to a variety of reasons (initialization, lack of priors for Bayesian neural networks, posterior collapse, optimization issues) we are not aware of a SVI variant that consistently achieves competitive accuracy as well as reliable uncertainty on large-scale problems (i.e. bigger than MNIST). Indeed, even after tremendous effort and incorporating a bag of tricks from the literature (careful initialization, empirical Bayes for the prior standard deviation), we were unable to get competitive accuracy on e.g. CIFAR. We will provide more details and analysis in the paper. Apart from optimization issues, another possibility is that SVI works well for datasets with well-specified models (e.g. models on MNIST achieve more than 99% accuracy), and underperforms on datasets where model is mis-specified. See also "Bootstrap prediction and Bayesian prediction under misspecified models" (Fushiki 2005) for theoretical arguments. To test this hypothesis, we will add an experiment on MNIST where we increase model mis-specification by reducing capacity (and accuracy) and compare the performance of ensemble and SVI.

**R1:** *"some of the baselines, seem to not be properly tuned... For example, for ImageNet and Skew Intensity 1, LL-SVI and Dropout underperform compared to Vanilla, ..."* In Sec A.7 we detail the tuning methodology that was used across models. However, R1 brings up a good point w.r.t. model capacity. We fixed the size of the models to have an "apples to apples" comparison, but dropout may require larger capacity models. Note, LL-SVI was trained with the rest of the model (under the ELBO) and not post-hoc which may work better. We will include these experiments in the final paper.

**R2:** *"In my opinion, the significance is somewhat reduced because only neural network models are used"* We empathize and agree that other models may allow for more principled uncertainty. However, given the widespread use of deep networks, we believe that understanding and benchmarking the uncertainty of this class of models is important. We struggled to find alternatives that were competitive across these tasks. Nevertheless, scaling up other models is a priority for us and we trained a variational GP on raw MNIST (incl a variety of kernels) to address this. However, we found that they were not competitive in accuracy. We will add this (and on Criteo) to the camera ready.

*"Should one create one's own skew and OOD datasets? Are there any principles that are important to keep in mind while doing that? How hard will it be to use your eventually released code to do that?"* We hope these experiments will generalize, but we believe it should be quite easy to add new datasets and experiments to our code (which has now been released). We will add details of how to do this, along with recommendations, to the paper.

*"... Does more capacity mean better OOD calibration?"* We discuss the tradeoffs in terms of memory and computational complexity in the appendix section A.8. However, the question of whether the additional capacity of ensembling gives an advantage is an interesting one. We will add an experiment to the paper where we train and evaluate higher capacity networks for the dropout and vanilla methods to explore this.

*"details about ensemble"*: we used just a plain ensemble (no adversarial training, no heteroscedastic loss).

**R3:** *"Have the authors investigated the performance on different OOD datasets that are more / less similar to the source dataset...?"* We used the corruptions benchmark (Hendrycks and Dietterich, 2019) as it allows us to control the amount of similarity to source dataset, and see if consistent orderings are observed.

*"analysis / actionable takeaways."* This was limited due to space constraints, but we will add this to the supplement.

*"I found the experimental section somewhat disorganized ... would have made for a significantly better read."* Thanks for pointing this out. We will address the layout and presentation, particularly w.r.t. these figures, for the camera ready.

*"... While SVI does seem to have lower Brier scores with shift, accuracies don't appear to be any better – how was this observation made?"* Valid point. This wasn't well worded, but we meant specifically with regard to uncertainty as it's significantly better in Brier score but not accuracy - i.e. much better calibrated under significant shift. We will reword.

*"... What are these additional properties that ECE and entropy capture?"* Each proper scoring rule induces a calibration measure, see [Brocker, "Reliability, sufficiency, and the decomposition of proper scores", Quarterly Journal of the Royal Meteorological Society, 2009]. However, ECE is not the result of such decomposition and has no corresponding proper scoring rule; we instead chose to include ECE because it is popularly used. Each proper scoring rule is also associated with a corresponding entropy function and Shannon entropy is that for log probability [see Gneiting & Raftery, Journal of the American Statistical Association, 2007]. We will reword this and clarify our reasoning for the camera ready.

[Meta-Review · NeurIPS 2019]

The reviewers arrived at a consensus and recommend to accept this submission. Please incorporate the reviewers' suggested improvements into your camera ready version of the paper.